# RefRef: A Dataset and Benchmark for Reconstructing Refractive and Reflective Objects

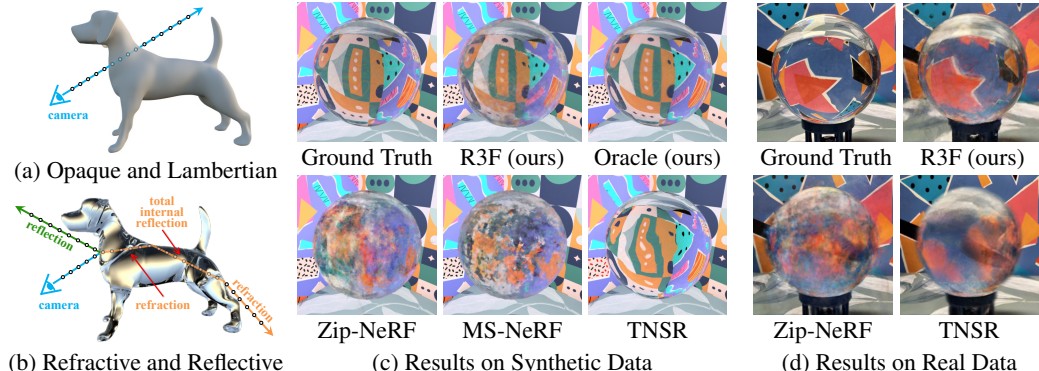

(a) Opaque and Lambertian

Ground Truth R3F (ours) Oracle (ours) Ground Truth R3F (ours)

(b) Refractive and Reflective

Zip-NeRF MS-NeRF TNSR Zip-NeRF TNSR

(c) Results on Synthetic Data (d) Results on Real Data

Figure 1: Comparison of light interactions with opaque Lambertian objects and refractive/reflective objects. (a) Opaque Lambertian objects allow modeling of light paths as linear. (b) Refractive and reflective objects lead to the curving and branching of light paths. (c–d) Novel view synthesis results on synthetic and real data. Our oracle performs best, while our R3F method loses some high-frequency detail and TNSR (Deng et al., 2024a) sacrifices geometric quality for visual crispness.

## Abstract

Modern 3D reconstruction and novel view synthesis approaches have demonstrated strong performance on scenes with opaque, non-refractive objects. However, most assume straight light paths and therefore cannot properly handle refractive and reflective materials. Moreover, datasets specialized for these effects are limited, stymieing efforts to evaluate performance and develop suitable techniques. In this work, we introduce the RefRef dataset for reconstructing scenes with refractive and reflective objects from posed images. Our dataset has 50 synthetic objects of varying complexity, from single-material convex shapes to multi-material non-convex shapes, each placed in three different background types, resulting in 150 scenes. A real scene that mirrors the synthetic setup is also provided for comparison. We propose an oracle method that, given the object geometry and refractive indices, calculates accurate light paths for neural rendering, and an approach based on this that avoids these assumptions. We benchmark these against several state-of-the-art methods and show that all methods lag significantly behind the oracle, highlighting the challenges of the task and dataset.

## 1 Introduction

Refractive and reflective objects, such as glass and water, pose significant challenges for 3D reconstruction and novel view synthesis due to the complex behavior of light as it passes through or reflects off these materials. Accurate modeling of these optical phenomena is essential for precise 3D reconstruction and photorealistic rendering. However, most existing neural radiance approaches (Mildenhall et al., 2021; Barron et al., 2022; Chen et al., 2022; Müller et al., 2022), while excelling at handling opaque Lambertian objects, struggle with refractive or reflective surfaces (Deng et al., 2024b). These methods typically assume that light travels in a straight path, which is valid for Lambertian objects, as illustrated in Figure 1a, but fails in scenarios with complex optical behavior. This

limitation is further compounded by the lack of suitable datasets focused on refractive and reflective properties, hindering the development and evaluation of more sophisticated methods.

To address these challenges, we introduce RefRef, a dataset and benchmark designed for the task of reconstructing scenes with complex refractive and reflective objects, as shown in Figure 1d. RefRef consists of 50 objects categorized based on their geometric and material complexity: single-material convex objects, single-material non-convex objects, and multi-material non-convex objects, where the materials have different colors, opacities, and refractive indices. Each object is rendered in three background settings: two bounded and one unbounded, resulting in 150 unique scenes with diverse geometries, material properties, and backgrounds. This provides a controlled environment for evaluating and developing 3D reconstruction methods that handle complex optical effects.

We also propose an oracle method that has access to the ground-truth geometry and refractive indices of refractive objects in the scene, allowing it to compute accurate light paths, as shown in Figure 1b. This approach provides a performance target for NeRF-based methods, showing how well they can perform if light paths are properly modeled. We then propose a relaxation of the oracle method—R3F—that circumvents its ground-truth requirements. Finally, we conduct an extensive evaluation of existing state-of-the-art methods, some of which are shown in Figure 1c, highlighting their short-comings in handling scenes with refractive and reflective properties. Our contributions are:

1. a dataset for 3D reconstruction of scenes with refractive and reflective objects;
2. an oracle method that models light paths using ground-truth geometry and refractive indices;
3. a method that relaxes these requirements by estimating and smoothing the object geometry; and
4. a benchmark evaluating state-of-the-art methods on this dataset, revealing the limits of existing approaches at handling complex optical phenomena.

## 2 RELATED WORK

**Neural 3D reconstruction.** Neural Radiance Field (NeRF) (Mildenhall et al., 2021) optimizes the parameters of a coordinate network to map from spatial position and viewpoint to color and density. To improve efficiency and accuracy, Zip-NeRF (Barron et al., 2023) combines a feature grid with anti-aliasing. Other approaches fit implicit geometry representations through volume rendering, including UNISURF (Oechsle et al., 2021), VolSDF (Yariv et al., 2021), and NeuS (Wang et al., 2021), which model 3D surfaces using occupancy or signed distance fields. Another line of approaches rasterizes 3D Gaussians (Kerbl et al., 2023; Szymanowicz et al., 2025), allowing for real-time rendering with high visual quality. However, while some methods have explored single reflections, including general NeRF-based approaches (Verbin et al., 2022b; Ma et al., 2024; Tiwary et al., 2023; Qiu et al., 2023; Han et al., 2024), as well as recent Gaussian splatting techniques (Jiang et al., 2024; Yang et al., 2024; Ye et al., 2024; Yao et al., 2024), none explicitly handle the curved light paths necessary for modeling refraction.

**Transparent object modeling.** Modeling refractive and reflective materials is a significant challenge due to the complexity of the light paths (Zhang et al., 2024; Li et al., 2020; Yin et al., 2023; Deng et al., 2024b; Wang et al., 2023; Gao et al., 2023; Qiu et al., 2023; Huang et al., 2025). Some approaches focus on light transport in dynamic water surfaces and underwater scenes (Xiong & Heidrich, 2021), while others reconstruct opaque objects in a single refractive medium (Tong et al., 2023; Zhan et al., 2023; Sun et al., 2024; Cassidy et al., 2020). Moenne-Loccoz et al. (2024) and Wu et al. (2025) perform ray tracing on volumetric Gaussian particles, enabling efficient simulation of secondary rays required for rendering phenomena such as reflection, refraction, and shadows. Li et al. (2020) assume known environment illumination and refractive indices, integrating rendering and cost volume layers to model reflection and refraction for precise point cloud reconstruction. Controlled experimental setups, such as using gray-coded patterns to determine ray–position correspondences, have improved surface reconstruction accuracy (Lyu et al., 2020; Wu et al., 2018; Li et al., 2023), but cannot be assumed in general. MS-NeRF (Yin et al., 2023) introduces multi-space feature fields to jointly model multiple subspaces, such as the real world and the world reflected in a mirror, giving it some capacity to model refraction. NEMTO (Wang et al., 2023) directly predicts the exiting light direction by treating the internal refraction and reflection processes as a black box and assuming an infinitely-distant background. As a result, it cannot handle the general reconstruction of posed images. Yoon & Lee (2024) use a visual hull method to approximate the object's shape and then consider two-bounce light paths. While this improves view synthesis results, it suffers

Table 1: Comparison of datasets by scenes $|\mathcal{S}|$, images $|\mathcal{I}|$, presence of refractive/reflective objects, ground-truth geometry, multi-material composition, bounded scenes, and tinted refractive materials.

| Datasets | $|\mathcal{S}|$ | $|\mathcal{I}|$ | Refr./Refl. | Geometry | Multi-mat. | Bounded | Tinted |
|---|---|---|---|---|---|---|---|
| DTU (Jensen et al., 2014) | 124 | 4.2k | ✗ | ✓ | ✓ | ✓ | ✗ |
| T & T (Knapitsch et al., 2017) | 7 | 88k | ✗ | ✓ | ✓ | ✗ | ✗ |
| ShapeNet (Chang et al., 2015) | 51k | 0 | ✗ | ✓ | ✓ | ✗ | ✗ |
| Omniobject3D (Wu et al., 2023) | 6k | 0 | ✗ | ✓ | ✓ | ✗ | ✗ |
| ObjaverseXL (Deitke et al., 2024) | 10M | 0 | ✗ | ✓ | ✓ | ✗ | ✗ |
| Shiny (Wizadwongsa et al., 2021) | 8 | 879 | ✓ | ✗ | ✗ | ✗ | ✗ |
| OpenMaterial (Dang et al., 2024) | 1k | 90k | ✓ | ✓ | ✗ | ✗ | ✗ |
| RefRef (Ours) | 150 | 45k | ✓ | ✓ | ✓ | ✓ | ✓ |

from voxelization artifacts. Ray Deformation Networks (Deng et al., 2024b) propose a deformation field to predict light bending in refractive objects, which works well for small levels of refraction and low-frequency shapes. TNSR (Deng et al., 2024a) builds on NeuS by integrating ray tracing and sphere tracing, improving view synthesis and geometry refinement; a similar strategy is taken by Gao et al. (2023). Overall, most methods assume only two refractions and one reflection, limiting their ability to model complex light interactions such as total internal reflection and multiple successive refractions, which frequently occur in real-world transparent and reflective objects.

**Datasets.** As shown in Table 1, several datasets have been widely adopted for 3D reconstruction tasks, each offering unique strengths for general scene understanding. The DTU dataset (Jensen et al., 2014) provides images across multiview setups. Tanks and Temples (Knapitsch et al., 2017) provides a benchmark for 3D reconstruction with complex scenes captured using high-precision scanners, suitable for evaluating methods in real-world settings. Additionally, the Objaverse-XL (Deitke et al., 2024), ShapeNet (Chang et al., 2015), and OmniObject3D (Wu et al., 2023) datasets extend the scale of the data but restrict the scope to object-centric reconstruction. These datasets contain too few refractive and reflective objects and so are not suitable for evaluating in this domain. Several datasets address the unique challenges posed by refractive and reflective objects. The Shiny dataset (Wizadwongsa et al., 2021) introduces complex view-dependent effects, such as rainbow reflections and refractions through glassware, designed to evaluate view synthesis under challenging conditions, however, it only includes 8 scenes, mostly with opaque objects. OpenMaterial (Dang et al., 2024) is a synthetic dataset that offers 295 distinct materials, but only 14% of its materials exhibit refractive properties with high transmittance, and it lacks multi-material or tinted objects, limiting its realism for scenes involving complex object interactions. Unlike the above datasets, our RefRef dataset is designed to benchmark 3D reconstruction methods for handling refractive and reflective objects. It contains 50 objects with varying materials, including single-material and multi-material objects with different refractive indices and tints, providing a wide range of challenges.

## 3 A REFRACTIVE–REFLECTIVE OBJECT DATASET

**Dataset structure.** The dataset contains 50 synthetic objects, each placed in three different backgrounds, resulting in 150 scenes, as well as one similar real scene to test generalization. The objects are categorized into three categories based on their geometry (convex or non-convex) and refractive material count (single or multiple) to evaluate reconstruction methods at different difficulty levels.

1. **Single-material convex (27 scenes).** Objects with convex geometries, each composed of a single refractive material, such as transparent cubes, balls, cylinders, and pyramids.
2. **Single-material non-convex (60 scenes).** Objects with non-convex geometries, each composed of a single refractive material, such as animal sculptures, glass jars, light bulbs, and magnifiers.
3. **Multiple-materials non-convex (63 scenes).** Objects with non-convex geometries, each composed of multiple refractive materials, such as reed diffusers, a glass of wine, a cup of tea, and flasks filled with liquid chemicals.

**Scene backgrounds and cameras.** For each object, we generate three background environments to enhance variability in the rendered scenes: a cube background, a sphere background, and an HDR environment map. Each cube background scene is constructed by randomly selecting 6 images from a pool of 24 highly textured images; each sphere background is created by choosing 1 image from

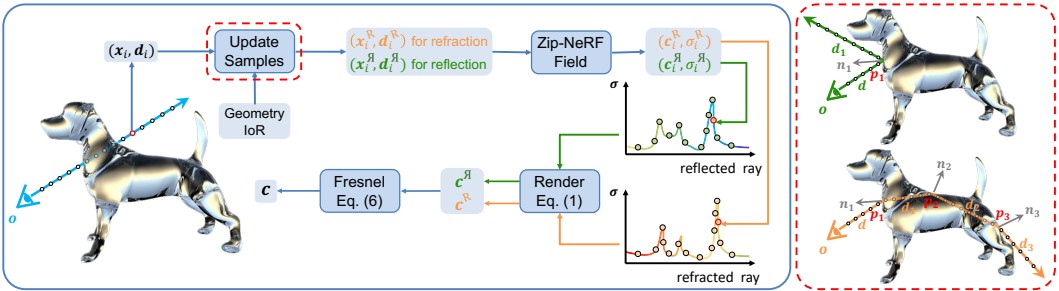

Figure 2: Overview of the oracle method. Starting with a straight ray (blue arrow), sample points are generated along its path. The scene's geometry and refractive index update the ray trajectory, as shown in the red dashed box. Refracted and reflected rays are processed separately, producing updated sample positions and directions. The Zip-NeRF field then predicts color $\mathbf{c}_i$ and density $\sigma_i$ for each sample. Using Eq. (1), the final color along each ray is rendered and combined via Eq. (6). Note: Zip-NeRF samples points in a conical spiral; we show only the centerline for clarity.

a set of 13; and the HDR environment map features an outdoor scene to provide realistic lighting conditions. This yields a total of 150 unique scenes. The images for each scene are subdivided into training, validation, and test sets. Each set consists of 100 images at a resolution of $800 \times 800$ pixels, accompanied by metadata, including camera positions, depth maps, 3D object models, and object masks. The camera viewpoints for the training and validation sets are randomly selected on a sphere centered around the object, ensuring diverse perspectives. For the test set, we employ a helical path, capturing 100 viewpoints by gradually ascending the camera position around the object.

**Rendering Process.** The dataset was rendered using Blender's Cycles path tracer (Blender, 2024). The renderer accounts for refraction, reflection, total internal reflection, and absorption, which are modeled using the Principled BSDF shader. More detailed settings are provided in Appendix A.3.

## 4 A VOLUME RENDERING ORACLE METHOD

Given a set of posed RGB images of a scene, NeRF-based (Mildenhall et al., 2021) methods typically estimate the geometry and appearance by learning a volumetric representation that can render images from novel views. However, most NeRF-based methods struggle with scenes involving refractive and reflective objects due to their assumption of straight light paths (Deng et al., 2024b). To evaluate how well a NeRF-based model can perform in reconstructing scenes with refractive and reflective objects, we present an oracle method that has access to the ground-truth geometry and refractive indices, as shown in Figure 2. The method assumes piecewise linear light paths (i.e., piecewise constant refractive indices), a single explicit reflection occurring at the first surface intersection, and a maximum of 10 refractions or total internal reflections. Note that NeRF-based view-dependent color prediction can allow for additional implicit reflections. For real refractive objects, these assumptions are not very restrictive.

### 4.1 NeRF PRELIMINARIES

The Neural Radiance Field (NeRF) method (Mildenhall et al., 2021) parametrizes a 3D scene as a radiance field coordinate network $\phi_\theta$, which maps a 5 DoF position and view direction $(\mathbf{x}_i, \mathbf{d})$ to volume density $\sigma_i$ and view-dependent color $\mathbf{c}_i$. The parameters $\theta$ of $\phi_\theta$ are optimized with respect to a photometric loss that compares renders of the neural field to the ground-truth images. To render an image, NeRF integrates the predicted colors $\mathbf{c}_i(\mathbf{x}_i, \mathbf{d})$ and densities $\sigma_i(\mathbf{x}_i)$ along each camera ray $\mathbf{r}(t) = \mathbf{o} + t\mathbf{d}$, where $\mathbf{o}$ and $\mathbf{d}$ denote the camera origin and direction, and $t$ ranges from the near plane $t_n$ to the far plane $t_f$. The rendered color $\mathbf{c}(\mathbf{r})$ of each pixel is approximated as the $w_i$-weighted sum of the colors $\mathbf{c}_i$ at $N$ sample points along the ray,

$$\mathbf{c}(\mathbf{r}) = \sum_{i=1}^{N} \underbrace{T_i(1 - \exp(-\sigma_i \Delta t_i))}_{w_i} \mathbf{c}_i, \tag{1}$$

where $T_i = \exp\left(-\sum_{j=1}^{i-1}\sigma_j \Delta t_j\right)$ denotes the accumulated transmittance up to sample $i$, and $\Delta t_i$ denotes the distance between consecutive samples.

## 4.2 Modeling Refractions and Reflections

This section outlines how the piecewise linear refracted and reflected light trajectories are computed, sampled, integrated along, and combined in our oracle method. It extends the robust Zip-NeRF (Barron et al., 2023) model to handle refraction and explicit reflection, as shown in Figure 2.

The oracle method represents light paths as piecewise linear functions parametrized by $K+1$ points $\{\mathbf{p}_i\}_{i=0}^K$ and unit direction vectors $\{\mathbf{d}_i\}_{i=0}^K$,

$$\mathbf{r}(t) = \sum_{i=0}^{K} [\![ t \in [\tau_i, \tau_{i+1}) ]\!] \left( \mathbf{p}_i + (t - \tau_i)\mathbf{d}_i \right),\tag{2}$$

where $[\![ \cdot ]\!]$ is an Iverson bracket, $\mathbf{p}_0 = \mathbf{o}$, $\mathbf{d}_0 = \mathbf{d}$ and the cumulative distance is given by

$$\tau_i = \begin{cases} 0 & \text{for } i = 0 \\ \infty & \text{for } i = K+1 \\ \sum_{j=1}^{i} \|\mathbf{p}_j - \mathbf{p}_{j-1}\| & \text{otherwise.} \end{cases}\tag{3}$$

The method considers two paths: a (multiple) refraction path $\mathbf{r}^{\mathfrak{R}}$ and a (single) reflection path $\mathbf{r}^{\mathfrak{R}}$. We next show how to compute the parameters for each.

**Refraction and reflection parameters.** For the refraction ray $\mathbf{r}^{\mathfrak{R}}$, given a position $\mathbf{p}_i$ and direction $\mathbf{d}_i$, the next intersection $\mathbf{p}_{i+1}$ with the refractive object is computed using ground-truth geometry. Let $\alpha_i = \nu_i/\nu_{i+1}$, $\beta_i = -\mathbf{d}_i^\mathsf{T}\mathbf{n}(\mathbf{p}_{i+1})$, $\gamma_i^2 = 1 - \alpha_i^2(1 - \beta_i^2)$, $\nu_i$ be the refractive index of the $i^{\text{th}}$ medium, and $\mathbf{n}(\mathbf{x})$ be the ground-truth unit normal vector at location $\mathbf{x}$. Then if $\gamma_i^2 \geqslant 0$, refraction will occur and the next direction $\mathbf{d}_{i+1}$ is computed using Snell's Law (Born & Wolf, 2013),

$$\mathbf{d}_{i+1} = \alpha_i \mathbf{d}_i + (\alpha_i \beta_i - \gamma_i)\,\mathbf{n}(\mathbf{p}_{i+1}).\tag{4}$$

Otherwise, total internal reflection will occur and the direction is given by the Law of Reflection,

$$\mathbf{d}_{i+1} = \mathbf{d}_i - 2(\mathbf{d}_i^\mathsf{T}\mathbf{n}(\mathbf{p}_{i+1}))\mathbf{n}(\mathbf{p}_{i+1}).\tag{5}$$

For the reflection ray $\mathbf{r}^{\mathfrak{R}}$, only the reflection at the first surface intersection is considered. The first reflected direction, $\mathbf{d}_1^{\mathfrak{R}}$, is given by Eq. (5). The proposed method does not explicitly model any other reflections (except for total internal reflections), because they are computationally expensive and have negligible impact in most situations.

**Sampling and rendering.** For neural rendering, our model samples points along the refraction and reflection paths using the proposal sampler from Mip-NeRF 360 (Barron et al., 2022), which first uniformly samples along the path and then uses the computed probability density function to concentrate samples in higher density regions. The neural field is queried at each sample location $\mathbf{x}_i$, using the corresponding direction vector $\mathbf{d}_i$ at that location (unlike standard NeRF that uses a constant direction $\mathbf{d}$). That is, we obtain $(\mathbf{c}_i^{\mathfrak{R}}, \sigma_i^{\mathfrak{R}}) = \phi(\mathbf{x}_i^{\mathfrak{R}}, \mathbf{d}_i^{\mathfrak{R}})$ and $(\mathbf{c}_i^{\mathfrak{R}}, \sigma_i^{\mathfrak{R}}) = \phi(\mathbf{x}_i^{\mathfrak{R}}, \mathbf{d}_i^{\mathfrak{R}})$ for all sample points on both paths. We then apply Eq. (1) to obtain the colors $\mathbf{c}^{\mathfrak{R}}$ and $\mathbf{c}^{\mathfrak{R}}$. The refractive and reflective color contributions are combined using the Fresnel equations (Hecht, 2012),

$$\mathbf{c}' = R(\mathbf{c}^{\mathfrak{R}} - \mathbf{c}^{\mathfrak{R}}) + \mathbf{c}^{\mathfrak{R}}, \quad \text{where} \quad R = \frac{1}{2}(R_p + R_s),\tag{6a}$$

$$R_p = \left(\frac{\nu_1 \beta_0 - \nu_0 \gamma_0}{\nu_1 \beta_0 + \nu_0 \gamma_0}\right)^2, \quad R_s = \left(\frac{\nu_0 \beta_0 - \nu_1 \gamma_0}{\nu_0 \beta_0 + \nu_1 \gamma_0}\right)^2,\tag{6b}$$

where $R_p$ and $R_s$ are the reflection coefficients for parallel and perpendicular polarized light; the color $\mathbf{c}'$ is converted to a non-linear sRGB space to obtain the predicted color $\hat{\mathbf{c}}$ (Verbin et al., 2022a).

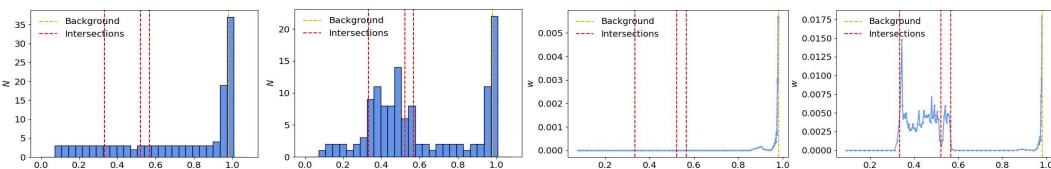

(a) $\mathcal{L}_{\text{dist}}^{\text{orig}}$ sample distribution (b) $\mathcal{L}_{\text{dist}}$ sample distribution (c) $\mathcal{L}_{\text{dist}}^{\text{orig}}$ weight distribution (d) $\mathcal{L}_{\text{dist}}$ weight distribution

Figure 3: Comparison of the original (Barron et al., 2022) distortion loss $\mathcal{L}_{\text{dist}}^{\text{orig}}$ and the translucency-corrected distortion loss $\mathcal{L}_{\text{dist}}$. (a)–(b) Distribution of sample points. (c)–(d) Distribution of weights.

### 4.3 OPTIMIZATION

The parameters $\theta$ of the coordinate network $\phi_\theta$ are optimized with respect to a photometric loss $\mathcal{L}_{\text{rgb}}$, an anti-aliased interlevel loss $\mathcal{L}_{\text{int}}$, and a modified distortion loss $\mathcal{L}_{\text{dist}}$. The per-pixel photometric loss is given by the mean squared color error,

$$\mathcal{L}_{\text{rgb}}(\hat{\mathbf{c}}, \mathbf{c}) = \frac{1}{3}\|\hat{\mathbf{c}} - \mathbf{c}\|_2^2, \tag{7}$$

where $\mathbf{c}$ is the ground-truth pixel color. The interlevel loss $\mathcal{L}_{\text{int}}$ (Barron et al., 2023) encourages consistency between the proposal network, which is used for quickly identifying important sampling regions, and the main network.

The distortion loss $\mathcal{L}_{\text{dist}}^{\text{orig}}$ introduced in Mip-NeRF 360 (Barron et al., 2022) encourages the weight distribution along a ray to coalesce and otherwise tend to zero. This leads to a preference for a single high-weight peak, that is, a single opaque surface. This is desirable for standard NeRF settings, since it reduces 'floaters' and background collapse. However, it is not applicable for translucent objects, where color contributions along a ray arise from both the translucent and opaque media. As shown in Figure 3 (left), the unmodified loss tends to reduce the density of translucent objects to zero, which has the side-effect of reducing samples within the object, due to proposal sampling. Thus, we propose a modified distortion loss that excludes samples within the refractive object, given by

$$\mathcal{L}_{\text{dist}}(\mathbf{s}, \mathbf{w}) = \sum_{i,j \in \mathcal{I}_{\text{out}}} w_i w_j \left| \frac{s_i + s_{i+1}}{2} - \frac{s_j + s_{j+1}}{2} \right| + \frac{1}{3} \sum_{i,j \in \mathcal{I}_{\text{out}}} w_i^2 (s_{i+1} - s_i), \tag{8}$$

where $s_i$ denotes the normalized ray distance (Barron et al., 2022) to the $i^{\text{th}}$ sample point, and $\mathcal{I}_{\text{out}}$ is the set of indices outside the refractive object. As shown in Figure 3 (right), after applying the corrected distortion loss, the model allocates more samples with higher weights within the refractive object, leading to more accurate reconstruction. The overall loss is then

$$\mathcal{L} = \mathcal{L}_{\text{rgb}} + \lambda_1 \mathcal{L}_{\text{int}} + \lambda_2 \mathcal{L}_{\text{dist}}, \tag{9}$$

where hyperparameters $\lambda_1$ and $\lambda_2$ weight the relative loss contributions and each loss term averages the per-pixel losses across the dataset.

## 5 R3F: A RELAXATION OF THE ORACLE METHOD

Given the previously described oracle method, the requirements of ground-truth object geometry and refractive index can be relaxed in a relatively straightforward manner. We name the resulting 3D reconstruction method R3F (Refractive–Reflective Radiance Field). The geometry of the refractive object is estimated using a modern variant of the visual hull algorithm (Laurentini, 1994), where posed object masks are given as inputs to the UNISURF implicit surface model (Oechsle et al., 2021). The object masks, if unavailable, may be accurately estimated using foreground–background segmentation (Kirillov et al., 2023). However, the resulting surfaces are insufficiently smooth for computing refracted light paths, since slight aberrations in the normal directions can cause large visual differences. To address this, we apply automatic post-processing to smooth and refine the surface, detailed in Appendix A.2. This provides the object geometry required by our oracle method. The refractive index, if unavailable, may be very precisely estimated using the approach outlined in TNSR (Deng et al., 2024a). While effective, R3F is limited to reconstructing the visual hull of the object, and so is not suitable for the multi-material category and objects with holes in the RefRef dataset, where internal structures need to be considered.

Table 2: Quantitative results on the RefRef test set. Metrics include view synthesis (PSNR, masked PSNR, SSIM, LPIPS) and geometry accuracy using distance mean absolute error (DMAE). DMAE is masked for natural backgrounds. Methods compared include NeuS (Wang et al., 2021), Splatfacto (Kerbl et al., 2023), Zip-NeRF (Barron et al., 2023), MS-NeRF (Yin et al., 2023), Ray Deformation Networks (Deng et al., 2024b), and RoseNeRF (Liang et al., 2024), along with our oracle and R3F.

| | Method | Cube Background | | | | | Sphere Background | | | | | Natural Background | | | | | All Backgrounds | | | | |
|---|---|---|---|---|---|---|---|---|---|---|---|---|---|---|---|---|---|---|---|---|---|
| | | PSNR ↑ | PSNR$_M$ ↑ | SSIM ↑ | LPIPS ↓ | DMAE ↓ | PSNR ↑ | PSNR$_M$ ↑ | SSIM ↑ | LPIPS ↓ | DMAE ↓ | PSNR ↑ | PSNR$_M$ ↑ | SSIM ↑ | LPIPS ↓ | DMAE$_M$ ↓ | PSNR ↑ | PSNR$_M$ ↑ | SSIM ↑ | LPIPS ↓ | DMAE ↓ |
| Convex single-mat. | NeuS | 18.48 | 14.94 | 0.66 | 0.19 | 1.20 | 20.77 | 14.88 | 0.60 | 0.16 | 1.81 | – | – | – | – | – | 19.62 | 14.91 | 0.63 | 0.18 | 1.50 |
| | Splatfacto | 22.32 | 14.90 | 0.87 | 0.11 | 14.56 | 21.67 | 15.00 | 0.88 | 0.16 | 33.41 | 11.88 | 13.05 | 0.62 | 0.71 | 1.17 | 18.88 | 14.36 | 0.80 | 0.31 | 16.96 |
| | Zip-NeRF | 22.16 | 14.88 | 0.86 | 0.13 | 0.10 | 22.95 | 15.70 | 0.91 | 0.13 | 0.22 | 33.04 | 26.52 | 0.94 | 0.10 | 0.08 | 25.78 | 18.75 | 0.90 | 0.12 | 0.17 |
| | TNSR | 19.30 | 11.92 | 0.84 | 0.12 | 0.90 | 18.62 | 11.66 | 0.85 | 0.17 | 1.62 | – | – | – | – | – | 18.96 | 11.79 | 0.85 | 0.14 | 1.26 |
| | MS-NeRF | 21.60 | 14.00 | 0.85 | 0.12 | 0.07 | 21.42 | 14.58 | 0.85 | 0.18 | 1.44 | 26.56 | 21.03 | 0.81 | 0.37 | 1.93 | 23.06 | 16.36 | 0.84 | 0.22 | 1.12 |
| | RayDef | 21.85 | 14.37 | 0.84 | 0.15 | 0.15 | 21.16 | 14.68 | 0.84 | 0.22 | 1.08 | 26.96 | 24.72 | 0.77 | 0.37 | 0.08 | 23.19 | 17.66 | 0.82 | 0.24 | 0.45 |
| | RoseNeRF | 20.18 | 14.02 | 0.82 | 0.32 | – | 22.33 | 15.22 | 0.88 | 0.14 | – | 23.54 | 26.35 | 0.77 | 0.51 | – | 22.67 | 17.94 | 0.81 | 0.31 | – |
| | R3F (Ours) | 23.55 | 16.49 | 0.86 | 0.12 | 0.08 | 25.08 | 18.10 | 0.90 | 0.12 | 0.18 | 30.91 | 24.17 | 0.93 | 0.13 | 0.01 | 26.51 | 19.58 | 0.90 | 0.12 | 0.09 |
| | Oracle (Ours) | 31.64 | 25.37 | 0.96 | 0.03 | 0.04 | 32.87 | 26.14 | 0.96 | 0.03 | 0.19 | 33.48 | 26.85 | 0.96 | 0.08 | 0.00 | 32.67 | 26.12 | 0.96 | 0.05 | 0.08 |
| Nonconvex single-mat. | NeuS | 19.11 | 14.86 | 0.67 | 0.13 | 1.17 | 20.72 | 13.21 | 0.62 | 0.13 | 1.97 | – | – | – | – | – | 19.92 | 14.03 | 0.64 | 0.13 | 1.57 |
| | Splatfacto | 24.40 | 16.02 | 0.88 | 0.07 | 9.42 | 22.07 | 15.30 | 0.85 | 0.17 | 26.30 | 10.71 | 11.64 | 0.51 | 0.76 | 0.97 | 19.17 | 14.36 | 0.75 | 0.33 | 12.73 |
| | Zip-NeRF | 24.15 | 15.87 | 0.88 | 0.08 | 0.09 | 24.37 | 15.91 | 0.89 | 0.09 | 0.20 | 27.17 | 19.10 | 0.89 | 0.14 | 0.15 | 25.23 | 16.96 | 0.89 | 0.10 | 0.18 |
| | TNSR | 18.92 | 11.25 | 0.83 | 0.14 | 1.31 | 19.38 | 11.85 | 0.83 | 0.16 | 1.62 | – | – | – | – | – | 19.22 | 11.57 | 0.83 | 0.15 | 1.47 |
| | MS-NeRF | 23.83 | 15.74 | 0.87 | 0.10 | 0.26 | 22.68 | 15.27 | 0.85 | 0.15 | 1.28 | 24.06 | 16.82 | 0.75 | 0.42 | 1.92 | 23.56 | 15.96 | 0.82 | 0.22 | 0.94 |
| | RayDef | 22.69 | 14.98 | 0.83 | 0.18 | 0.48 | 21.77 | 14.82 | 0.82 | 0.20 | 1.23 | 24.29 | 19.43 | 0.70 | 0.43 | 0.08 | 22.94 | 16.46 | 0.78 | 0.26 | 0.59 |
| | RoseNeRF | 22.83 | 14.97 | 0.81 | 0.15 | – | 22.49 | 15.12 | 0.85 | 0.13 | – | 23.76 | 17.15 | 0.72 | 0.58 | – | 23.02 | 16.21 | 0.79 | 0.24 | – |
| | R3F (Ours) | 23.17 | 15.12 | 0.87 | 0.10 | 0.11 | 23.27 | 15.03 | 0.88 | 0.11 | 0.25 | 26.54 | 18.66 | 0.87 | 0.20 | 0.03 | 24.33 | 16.27 | 0.87 | 0.14 | 0.13 |
| | Oracle (Ours) | 27.81 | 19.86 | 0.92 | 0.06 | 0.04 | 28.69 | 20.39 | 0.93 | 0.05 | 0.15 | 29.46 | 21.24 | 0.92 | 0.11 | 0.00 | 28.66 | 20.49 | 0.93 | 0.07 | 0.06 |
| Nonconvex multi-mat. | NeuS | 19.19 | 15.97 | 0.63 | 0.20 | 1.31 | 19.49 | 14.16 | 0.61 | 0.20 | 1.99 | – | – | – | – | – | 19.35 | 15.11 | 0.62 | 0.19 | 1.62 |
| | Splatfacto | 24.70 | 17.72 | 0.86 | 0.10 | 11.03 | 24.53 | 17.95 | 0.89 | 0.11 | 19.93 | 10.60 | 10.55 | 0.54 | 0.76 | 1.13 | 20.10 | 15.53 | 0.76 | 0.32 | 10.61 |
| | Zip-NeRF | 25.61 | 18.05 | 0.88 | 0.09 | 0.11 | 26.10 | 18.36 | 0.90 | 0.09 | 0.24 | 29.55 | 22.58 | 0.91 | 0.14 | 0.19 | 27.09 | 19.66 | 0.89 | 0.11 | 0.21 |
| | TNSR | 17.66 | 10.06 | 0.81 | 0.16 | 1.19 | 17.69 | 10.61 | 0.80 | 0.21 | 2.20 | – | – | – | – | – | 17.94 | 10.44 | 0.82 | 0.17 | 1.62 |
| | MS-NeRF | 24.93 | 17.89 | 0.85 | 0.12 | 0.39 | 20.94 | 15.27 | 0.78 | 0.27 | 2.56 | 25.99 | 20.36 | 0.78 | 0.40 | 0.19 | 23.64 | 16.84 | 0.82 | 0.24 | 1.02 |
| | RayDef | 23.94 | 16.66 | 0.84 | 0.14 | 0.34 | 21.10 | 15.56 | 0.80 | 0.27 | 1.60 | 24.85 | 22.09 | 0.72 | 0.43 | 0.12 | 23.29 | 18.12 | 0.78 | 0.28 | 0.69 |
| | RoseNeRF | 23.16 | 15.94 | 0.80 | 0.16 | – | 20.20 | 16.56 | 0.78 | 0.29 | – | 22.72 | 21.91 | 0.73 | 0.55 | – | 22.78 | 18.27 | 0.75 | 0.30 | – |
| | R3F (Ours) | 23.08 | 15.76 | 0.85 | 0.13 | 0.15 | 23.26 | 15.78 | 0.87 | 0.14 | 0.26 | 26.94 | 20.02 | 0.86 | 0.22 | 0.05 | 24.43 | 17.19 | 0.86 | 0.16 | 0.15 |
| | Oracle (Ours) | 27.45 | 20.02 | 0.91 | 0.08 | 0.04 | 27.76 | 20.05 | 0.92 | 0.08 | 0.14 | 30.82 | 23.61 | 0.92 | 0.13 | 0.00 | 28.67 | 21.23 | 0.92 | 0.10 | 0.06 |
| Entire dataset | NeuS | 19.15 | 15.33 | 0.65 | 0.17 | 1.23 | 20.15 | 13.97 | 0.61 | 0.16 | 1.93 | – | – | – | – | – | 19.62 | 14.64 | 0.63 | 0.19 | 1.62 |
| | Splatfacto | 24.26 | 16.69 | 0.87 | 0.09 | 10.96 | 23.03 | 16.36 | 0.87 | 0.14 | 24.90 | 10.86 | 11.39 | 0.54 | 0.75 | 1.08 | 19.53 | 14.87 | 0.77 | 0.32 | 12.55 |
| | Zip-NeRF | 24.41 | 16.60 | 0.88 | 0.09 | 0.10 | 24.84 | 16.90 | 0.91 | 0.10 | 0.22 | 29.15 | 21.80 | 0.91 | 0.13 | 0.16 | 26.11 | 18.41 | 0.89 | 0.11 | 0.20 |
| | TNSR | 18.78 | 11.02 | 0.83 | 0.18 | 1.10 | 18.51 | 11.28 | 0.82 | 0.18 | 1.87 | – | – | – | – | – | 18.64 | 11.14 | 0.83 | 0.16 | 1.49 |
| | MS-NeRF | 23.98 | 16.43 | 0.86 | 0.11 | 0.22 | 21.72 | 15.15 | 0.82 | 0.21 | 1.84 | 25.32 | 19.07 | 0.77 | 0.40 | 1.93 | 23.64 | 16.84 | 0.82 | 0.24 | 1.02 |
| | RayDef | 23.07 | 15.61 | 0.83 | 0.15 | 0.37 | 21.37 | 15.11 | 0.81 | 0.23 | 1.36 | 24.96 | 21.43 | 0.72 | 0.42 | 0.10 | 23.13 | 17.38 | 0.79 | 0.27 | 0.61 |
| | RoseNeRF | 22.49 | 15.07 | 0.83 | 0.18 | – | 21.50 | 15.74 | 0.83 | 0.20 | – | 23.28 | 20.81 | 0.73 | 0.55 | – | 22.86 | 17.39 | 0.78 | 0.28 | – |
| | R3F (Ours) | 23.19 | 15.61 | 0.86 | 0.12 | 0.12 | 23.56 | 15.85 | 0.88 | 0.12 | 0.25 | 27.43 | 20.14 | 0.88 | 0.20 | 0.03 | 24.73 | 17.20 | 0.87 | 0.15 | 0.13 |
| | Oracle (Ours) | 28.28 | 20.82 | 0.92 | 0.06 | 0.04 | 29.05 | 21.28 | 0.93 | 0.06 | 0.15 | 30.69 | 23.17 | 0.93 | 0.11 | 0.00 | 29.34 | 21.75 | 0.93 | 0.08 | 0.07 |

## 6 EXPERIMENTS

**Baselines and prior work.** Alongside the oracle and R3F methods, we evaluate several state-of-the-art approaches: Neural Surface Reconstruction (NeuS) (Wang et al., 2021), Splatfacto (Kerbl et al., 2023), Zip-NeRF (Barron et al., 2023), Multi-Space Neural Radiance Fields (MS-NeRF) (Yin et al., 2023), Ray Deformation Networks (RayDef) (Deng et al., 2024b), Transparent Neural Surface Refinement (TNSR) (Deng et al., 2024a), and RoseNeRF (Liang et al., 2024). The first three use the standard straight line light path assumption, while the others model refraction and reflection.

**Metrics.** To assess novel view quality, we report the peak signal-to-noise ratio (PSNR), the masked PSNR (PSNR$_M$) that excludes background to focus on refractive objects, the structural similarity index measure (SSIM), and the learned perceptual image patch similarity (LPIPS), which is more indicative of differences to the human eye than PSNR, especially with respect to blur. To assess the geometric fidelity, we report the distance mean absolute error (DMAE) between the estimated and ground-truth distance maps. Importantly, standard weighted-sum distance rendering techniques (Mildenhall et al., 2021) cannot be used for translucent objects, since weights are distributed across translucent and opaque media. Instead, we take the median distance of the weight samples along each ray, as a robust estimate of the distance to the nearest (potentially translucent) surface.

**Implementation details.** Oracle and R3F both extend Zip-NeRF (Barron et al., 2023) to allow piece-wise linear light paths and explicit reflection. For bounded scenes, we set near plane $t_n=0.05$, far

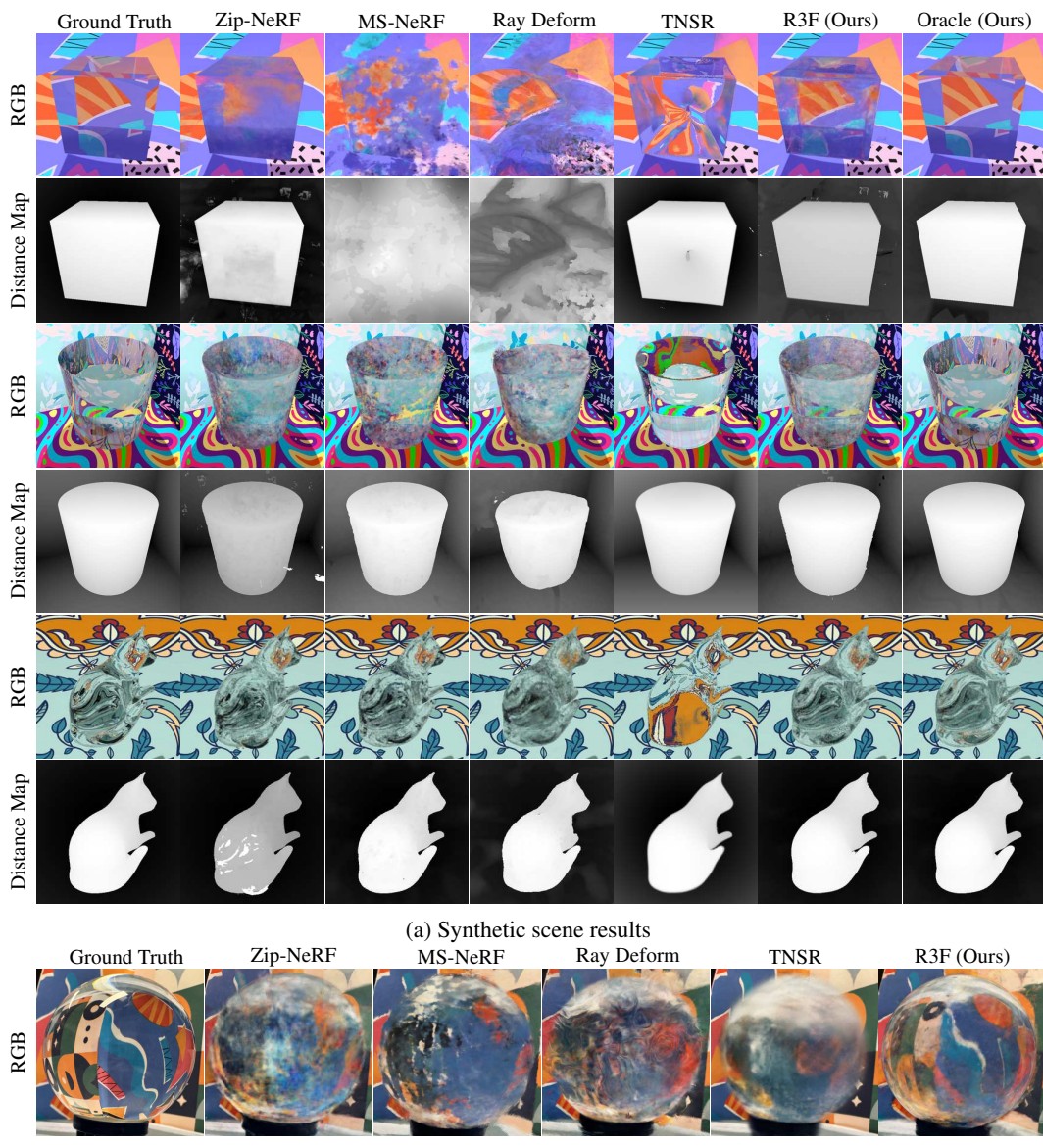

(a) Synthetic scene results

(b) Real scene results

Figure 4: Qualitative results on (a) synthetic and (b) real scenes. Our R3F and Oracle methods outperform Zip-NeRF (Barron et al., 2023), MS-NeRF (Yin et al., 2023), Ray Deformation Networks (Deng et al., 2024b), and TNSR (Deng et al., 2024a), especially in scenes with multiple refractions and total internal reflection, where other methods often fail.

plane $t_f=15$, and distortion loss weight $0.01$. For unbounded scenes, we use $t_f=1000$ and a contraction warp function. Models are trained for 25k iterations on a single A6000 GPU with batch size $4096$. Further implementation details are provided in Appendix A.2.

**Quantitative and qualitative results.** The quantitative results are reported in Table 2 for the three data subsets. R3F performs strongly in the single-material convex object category, outperforming all other methods except the oracle, which receives privileged information. However, it struggles with handling objects with concavities; a consequence of its reliance on a variant of the visual hull algorithm. The other methods perform reasonably well on simple scenes (e.g., convex objects, natural environment map backgrounds), but perform significantly worse for harder objects and background types. The object geometry is particularly poorly estimated, showing that the models are taking shortcuts either by deforming the geometry to fit the refracted appearance or by placing

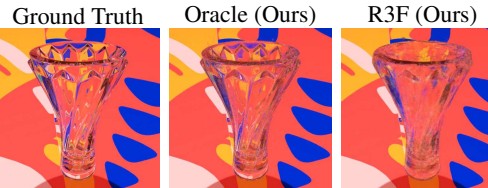

Ground Truth      Oracle (Ours)      R3F (Ours)

Figure 5: Failure case of the oracle and R3F methods. The oracle method, despite access to ground-truth geometry and refractive indices, struggles to model the vase's uneven surface. Meanwhile, R3F treats the vase as solid, causing appearance distortions near the top.

Table 3: Ablation study of the oracle method on single-material convex objects (cube background). Ablated components are distortion loss correction, first surface reflection, total internal reflection, and Zip-NeRF (Barron et al., 2023) backbone (substituting Nerfacto (Tancik et al., 2023)).

| Method | PSNR↑ | PSNR$_\text{M}$↑ | SSIM↑ $\times 10^{-2}$ | LPIPS↓ $\times 10^{-2}$ | DMAE↓ $\times 10^{-2}$ |
|---|---|---|---|---|---|
| Oracle | **31.64** | **25.37** | **96.13** | **2.76** | 4.38 |
| w/o corrected $\mathcal{L}_\text{dist}$ | 31.59 | 25.31 | 96.12 | 2.78 | 4.07 |
| w/o Я | 29.65 | 22.92 | 94.69 | 3.67 | **2.97** |
| w/o TIR | 25.85 | 19.27 | 90.84 | 8.38 | 15.86 |
| w/o Zip-NeRF | 24.68 | 21.45 | 87.79 | 11.63 | 27.42 |

floaters around objects. Moreover, R3F performs slightly worse than Zip-NeRF in the non-convex and multi-material categories. These subsets contain many thin glass objects where the straight-ray assumption, as made by Zip-NeRF, is reasonable. In contrast, R3F models these objects as solid volumes and bend s rays more than appropriate, resulting in higher errors. This reveals both a limitation of R3F and the overall difficulty of refractive reconstruction. Qualitative results for both synthetic and real scenes are presented in Figure 4. A clear performance gap is observed between our methods and existing approaches. In scenes involving multiple refractions and total internal reflections (TIR), most methods produce blurry or incorrect outputs. In contrast, the oracle and R3F methods capture these refracted background patterns more accurately, although R3F may exhibit artifacts due to inaccuracies in geometry estimation. Additional qualitative results are provided in Appendix A.6.

Furthermore, both the oracle method and R3F exhibit failure cases, particularly when reconstructing objects with complex structures, as shown in Figure 5. The oracle method, with access to ground-truth geometry, can capture fine details such as holes, but still struggles to model highly uneven surfaces. While R3F relies on a variant of the visual hull algorithm (Laurentini, 1994), which prevents it from modeling internal cavities, resulting in solid geometries and rendering artifacts. These examples highlight the difficulty of handling objects with hidden or internal structures, where accurate geometry estimation is critical. They also show that modeling refractive objects is challenging even with ground-truth data, pointing to the need for more robust methods.

**Ablation study.** An ablation study is presented in Table 3, comparing the full oracle method with four ablated versions: without the corrected distortion loss, first surface reflection, total internal reflection (TIR), and replacing the Zip-NeRF (Barron et al., 2023) backbone with Nerfacto (Tancik et al., 2023). The full oracle method accurately models both reflections and TIR, producing highly detailed renderings. In contrast, removing the corrected distortion loss reduces sample points within refractive objects, as illustrated in Figure 3. Omitting the first surface reflection leads to the loss of subtle reflective details, while disabling TIR results in missing critical light interactions within refractive objects, and replacing the backbone results in lower performance. These results highlight the importance of each component, as removing any of them leads to a clear performance drop.

# 7 CONCLUSION

We have presented the RefRef dataset for 3D reconstruction and novel view synthesis of scenes containing refractive and reflective objects. To establish a performance target, we introduce an oracle method based on ground-truth geometry and refractive indices, as well as a more practical alternative, R3F, that relaxes these assumptions. Benchmarking state-of-the-art methods revealed significant performance gaps, even for models explicitly designed to handle nonlinear light paths. More surprisingly, the oracle method exhibits several limitations despite its fairly mild assumptions (a maximum of ten bends, a single explicit reflection). This highlights the high sensitivity of light transport to geometric inaccuracies—small errors in surface normals can cause large deviations in ray paths. These results point to the need for new reconstruction methods that can more reliably account for complex light interactions in transparent and reflective scenes.

## ETHICS STATEMENT

Our work introduces a dataset and benchmark designed for reconstructing scenes containing refractive and reflective objects. While this addresses a known limitation in current 3D reconstruction and novel view synthesis methods, it carries both potential benefits and risks for society.

**Potential Positive Societal Impacts.** This work contributes to the advancement of computer vision research by enabling the development and evaluation of methods that can better model complex light interactions such as refraction and reflection. These capabilities are essential for accurately reconstructing scenes with non-Lambertian materials. Furthermore, the improvements in reconstruction quality have potential applications in robotics, autonomous navigation, and augmented or virtual reality, where reliable perception in complex environments is important. For example, a reconstruction method that fails to correctly model the 3D structure of a plastic object on a road may cause an accident for an autonomous driving system.

**Potential Negative Societal Impacts.** At the same time, the ability to reconstruct scenes containing reflective or transparent objects with greater fidelity could be misused for surveillance or the unauthorized reconstruction of private spaces, raising privacy concerns. In addition, reliance on large datasets for training and evaluation can lead to high computational costs. This contributes to increased energy consumption and environmental impact.

## REPRODUCIBILITY STATEMENT

We have made every effort to ensure reproducibility of our results. All training procedures, hyperparameters, and loss functions are fully described in the main paper and Appendix. The dataset, source files, rendering scripts, and algorithm code have all been fully implemented, organized, and tested, and will be made publicly available in accordance with ICLR's policy.

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

# A APPENDIX

## A.1 ADDITIONAL METHODOLOGICAL DETAILS

In this section, we provide additional details on our sampling strategy and how updated sample points are processed for density and color prediction in our oracle method. These extensions complement the core concepts outlined in the main paper and focus on implementation-specific details.

**Sampling Strategy.** We begin by generating an initial straight ray $\mathbf{r}(t) = \mathbf{o} + t\mathbf{d}$, which is then split into two paths: ta refraction path $\mathbf{r}^{\mathcal{R}}$ and a reflection path $\mathbf{r}^{\mathcal{A}}$. To sample along these paths, we adopt the proposal sampling strategy introduced in ZipNeRF (Barron et al., 2023), which builds upon the hierarchical sampling approach of Mip-NeRF 360 (Barron et al., 2022). ZipNeRF samples inside a cone following a spiral path, we update the center line of the cone while preserving the sampling pattern. It starts with a uniform sampler that generates $N$ sample points $\mathbf{x}_i$ along the initial straight ray. These points are then updated, resulting in new positions $\mathbf{x}_i^{\mathcal{R}}$ and $\mathbf{x}_i^{\mathcal{A}}$ with corresponding direction vectors $\mathbf{d}_i^{\mathcal{R}}$ and $\mathbf{d}_i^{\mathcal{A}}$, which follow curved paths for refractions and reflections, respectively. Next, the density $\sigma_i$ and weight $w_i$ of each sample point along these curved paths are computed. These weights, which represent the contribution of each sample, are further passed through a probability density function sampler to concentrate samples in higher density regions to enhance the accuracy of rendering.

**Processing Updated Sample Points.** After each sampling stage, the updated sample point positions $\mathbf{x}_i^{\mathcal{R}}$ and $\mathbf{x}_i^{\mathcal{A}}$ are processed through a spatial encoding function $\xi$ for efficient representation. These encoded spatial points are then fed into a multi-layer perceptron (MLP) $f_\theta$, which predicts the scene density $\sigma_i$. To obtain the predicted color at each 3D position, another MLP $g_\theta$ processes the refined viewing directions $\mathbf{d}_i^{\mathcal{R}}$ and $\mathbf{d}_i^{\mathcal{A}}$, encoded using spherical harmonics $\psi$, alongside the computed density $\sigma_i$. This yields the emitted color $\mathbf{c}_i = (R, G, B)$ for each sample point.

## A.2 FURTHER IMPLEMENTATION DETAILS

For optimizing the oracle method, we use an Adam optimizer with an initial learning rate of $8 \times 10^{-3}$, $\epsilon = 1 \times 10^{-15}$, and an exponential decay to $1 \times 10^{-3}$ over $2.5 \times 10^4$ steps, with a 1000-step warm-up. Scene contraction is applied for HDR environment map backgrounds.

For R3F, geometry post-processing begins with applying a convex hull to all meshes. We identify and remove floaters by computing the maximum convex-to-original vertex distance $\max_i(d_i)$. If this distance exceeds a threshold, we apply the Remove Isolated Pieces filter (90% diameter) in MeshLab (Cignoni et al., 2008) to eliminate disconnected or spurious components; otherwise, we retain the convex hull as the final mesh, assuming it sufficiently represents the original shape. In Blender, we further refine the meshes to improve smoothness and geometric fidelity. Specifically, we apply a bevel operation (Zorin et al., 1996) ($0.015m$, 3 segments) to round sharp edges, perform smoothing (Desbrun et al., 1999) (factor $= 1.0$, repeat $= 100$), and then utilize remeshing (Kobbelt et al., 2001) (smooth mode, octree depth $= 8$, scale $= 0.9$, threshold $= 1.0$).

## A.3 RENDERING DETAILS

We render all synthetic scenes using the Cycles path tracer in Blender (Blender, 2024), which employs Monte Carlo integration to solve the rendering equation using BSDFs (Bidirectional Scattering Distribution Functions) (Kajiya, 1986). Light transport is approximated by stochastically tracing paths of light rays and recursively evaluating their interactions with surfaces and transmissive materials:

$$L_o(x, \omega_o) = L_e(x, \omega_o) + \int_\Omega f_s(x, \omega_i, \omega_o) \, L_i(x, \omega_i) \, (\omega_i \cdot n) \, d\omega_i,$$

where $L_o(x, \omega_o)$ is the radiance leaving point $x$ in direction $\omega_o$, $L_e$ is the emitted radiance, $f_s$ is the BSDF, $L_i$ is the incoming radiance from direction $\omega_i$, $n$ is the surface normal at $x$, and $\Omega$ is the hemisphere of incoming directions. The dot product $\omega_i \cdot n$ accounts for the cosine falloff of incoming light. Unlike the BRDF, the BSDF $f_s$ accounts for both surface reflection and transmission, which is necessary for modeling materials such as glass and transparent plastics used in our scenes.

Table 4: Source and license information for objects used in our dataset. Objects are obtained from various sources: BlenderKit (`https://www.blenderkit.com/`), CGTrader (`https://www.cgtrader.com/`), Free3D (`https://free3d.com/`), Keenan's 3D Model Repository (Crane et al., 2013)(`https://www.cs.cmu.edu/~kmcrane/Projects/ModelRepository/`), and custom creations. All externally sourced objects were materially modified for our requirements.

| Source | License | Objects |
|---|---|---|
| BlenderKit | Royalty Free | cat, diamond, fox, man sculpture, sleeping dragon, candle holder, cola bottle, crystal vase, demijohn vase, flower vase, Korken jar, Vardagen jar, wisolt kettle, magnifier, plastic bottle, reed diffuser, skull bottle, teacup, teapot, water pitcher, wine glass, light bulb, perfume red, perfume yellow, star-shaped bottle, household item set, ampoule, beaker, conical flask, vial, lab equipment set |
| BlenderKit | CC0 | generic sculpture, woman sculpture |
| Blender | GPL | monkey, ball, coloured ball, cube, coloured cube, cylinder, coloured cylinder, pyramid, coloured pyramid, torus, coloured torus |
| CGTrader | Royalty Free | graduated cylinder, test tube, round bottom flask |
| Free3D | Royalty Free | dog |
| Keenan's | CC0 | cow |
| Authors | CC0 | syringe |

We model the scenes using physically realistic materials provided by Blender's Principled BSDF shader, specifying material parameters for glass, metal, and dielectric plastics. Object placements, lighting, and camera positions are randomized to encourage variation across samples. To ensure rendering quality:

- Adaptive sampling (Bucher, 1988) is enabled to improve rendering efficiency by dynamically adjusting the number of samples per pixel based on estimated noise. Pixels with low variance receive fewer samples, while high-frequency regions are sampled more densely. We set the noise threshold to 0.05, which defines the per-pixel error tolerance for terminating sampling. The minimum number of samples is set to 0, allowing Cycles to automatically determine this value from the threshold. The maximum number of samples per pixel is capped at 600.
- The maximum number of light bounces is set to 12 to support complex global illumination. This includes up to 4 diffuse bounces for soft indirect light, 4 glossy bounces for specular reflections, 12 transmission bounces to handle multiple refractive interfaces, and 8 transparent bounces for rays passing through alpha-masked geometry. Volume scattering is disabled (volume bounces = 0) since no participating media are present in our scenes.
- To reduce bright noise artifacts ("fireflies"), we clamp both direct and indirect light contributions to 1.0. We also enable reflective and refractive caustics to preserve high-frequency light transport effects such as focused reflections or refractions.
- The Glossy Filter is set to 0.0 to retain sharp caustic and highlight details without blurring.

## A.4 LICENSES

Detailed information on the sources and licensing of objects in our dataset is provided in Table 4, covering both objects created by our team and those sourced from online repositories with specific licensing terms. For sourced objects, we made material modifications where necessary to better align with the dataset's requirements.

Table 5: Quantitative results on the top-5 test views farthest from any training view in camera pose space. We report view synthesis metrics (PSNR, masked PSNR, SSIM, LPIPS) and geometry accuracy using the distance root mean square error (DRMSE).

| Method | PSNR↑ | PSNR$_M$↑ | SSIM↑ | LPIPS↓ | DRMSE↓ |
|---|---|---|---|---|---|
| ZipNeRF (Barron et al., 2023) | 25.37 | 17.81 | 0.89 | 0.12 | 1.21 |
| MS-NeRF (Yin et al., 2023) | 22.08 | 16.14 | 0.78 | 0.29 | 0.94 |
| RayDef (Deng et al., 2024b) | 19.99 | 16.28 | 0.72 | 0.38 | 1.98 |
| R3F (Ours) | 24.63 | 17.17 | 0.87 | 0.16 | 0.88 |
| Oracle (Ours) | **28.44** | **21.06** | **0.92** | **0.09** | **0.00** |

## A.5 Hyperparameters

In this section, we detail the hyperparameter settings used for training the models evaluated in our experiments. The configurations, including learning rates, optimizers, and scheduler settings, were carefully chosen to ensure stable convergence and performance across different methods. Specific hyperparameter choices for each evaluated approach are described below.

For MS-NeRF (Yin et al., 2023) and RoseNeRF (Liang et al., 2024), both the proposal networks and the field optimizer utilize an Adam optimizer with an initial learning rate of $4 \times 10^{-3}$, $\epsilon = 1 \times 10^{-15}$, and an exponential decay scheduler that reduces the learning rate to $1 \times 10^{-4}$ over $2 \times 10^{5}$ steps. For Zip-NeRF (Barron et al., 2023), the model optimizer utilizes a default configuration with an Adam optimizer set to an initial learning rate of $8 \times 10^{-3}$, $\epsilon = 1 \times 10^{-15}$, and an exponential decay scheduler reducing the learning rate to $1 \times 10^{-3}$ over $2.5 \times 10^{4}$ steps, with 1000 warm-up steps. For Ray Deformation Network (Deng et al., 2024b), both the proposal networks and the field optimizer utilize an Adam optimizer with an initial learning rate of $2 \times 10^{-3}$, $\epsilon = 1 \times 10^{-15}$, and an exponential decay scheduler reducing the learning rate to $1 \times 10^{-4}$ over $1 \times 10^{5}$ steps. For NeuS (Wang et al., 2021) and TNSR (Deng et al., 2024a), the proposal networks utilize a default configuration with an Adam optimizer set to an initial learning rate of $1 \times 10^{-2}$, $\epsilon = 1 \times 10^{-15}$, and a multi-step decay scheduler reducing the learning rate by a factor $\gamma = 0.33$ every milestone over $2 \times 10^{4}$ steps. The milestones are at 10000, 15000, and 18000 steps, respectively. For the field optimizer, we use an initial learning rate of $5 \times 10^{-4}$, and a cosine decay scheduler with 500 warm-up steps and learning rate peak value set to $5 \times 10^{-2}$. We have also included a 3D Gaussian splatting approach, Splatfacto (Kerbl et al., 2023), which is implemented in nerfstudio. For Splatfacto, we used the default configurations in nerfstudio.

## A.6 Extended Experimental Results

In this section, we compare more results of the oracle and R3F methods with existing state-of-the-art methods on our RefRef test set. We present comparative qualitative results on objects placed in patterned cube and patterned sphere backgrounds in Figure 6, and HDR environment map backgrounds in Figure 7. The patterned backgrounds are more challenging due to the presence of complex textures, where most methods struggle, especially in handling multiple refractions and total internal reflection. These methods are unable to reconstruct the highly detailed patterns, either blurry or incorrect. In contrast, the oracle and R3F methods are better able to capture these complex light interactions. The HDR environment map background is relatively easier, and most methods perform well on simple geometries. However, their performance fluctuates significantly on more complex shapes. In both settings, our approaches consistently produce more accurate and stable results, especially in scenes dominated by refractive components.

We also present further quantitative comparison in Table 5 on the top-5 test views farthest from any training view in camera pose space. While ZipNeRF (Barron et al., 2023) achieves competitive view synthesis metrics on the full dataset, its performance drops on this challenging subset, particularly in geometry accuracy, where it is notably worse than the oracle and R3F methods. This suggests that ZipNeRF may be improving appearance metrics by either deforming the geometry or introducing floaters around the object to compensate for refractive effects, rather than accurately modeling the physical light transport. In contrast, our R3F maintains stable performance across both the full dataset and this challenging subset, with only subtle differences in these metrics, demonstrating its robustness to viewpoint variations. The oracle method, as expected, achieves perfect geometry

Table 6: Quantitative results on the real scene test views. Metrics include PSNR, masked PSNR, SSIM, and LPIPS.

| Method | PSNR↑ | PSNR$_M$↑ | SSIM↑ | LPIPS↓ |
|---|---|---|---|---|
| ZipNeRF (Barron et al., 2023) | 19.70 | 16.46 | 0.79 | 0.34 |
| MS-NeRF (Yin et al., 2023) | 16.76 | 13.66 | 0.71 | 0.49 |
| RayDef (Deng et al., 2024b) | **20.11** | 16.45 | 0.74 | 0.42 |
| TNSR (Deng et al., 2024a) | 19.87 | 15.14 | **0.82** | **0.33** |
| R3F (Ours) | 18.98 | **17.30** | 0.70 | 0.43 |

reconstruction and superior rendering quality, further validating our theoretical framework. These results highlight the limitations of prior work in handling challenging refractive scenes under novel viewpoints and further demonstrate the robustness of our approaches.

Moreover, we evaluate several methods on the real scene to test generalization beyond synthetic data, with quantitative results reported in Table 6. ZipNeRF (Barron et al., 2023) and TNSR (Deng et al., 2024a) achieve higher SSIM and lower LPIPS, indicating they preserve overall structural similarity and perceptual quality, but they struggle under heavy refractions, as seen in their lower masked PSNR. Ray Deformation Networks (Deng et al., 2024b) obtains the highest overall PSNR, yet its performance drops considerably when evaluating only the refractive regions. In contrast, our R3F method achieves the strongest masked PSNR, demonstrating accurate reconstruction of the refractive object, which is the main challenge in real scenes. However, R3F produces minor artifacts in background regions, often appearing as blurred floaters. This behavior is likely due to the fact that R3F relies on estimated object geometry, and in real scenes the object's pose and size may deviate from the true values more than in synthetic data. Even small deviations in geometry or pose can cause large discrepancies in refracted light paths, leading the model to introduce floaters or blurred contributions around the foreground object to compensate. Despite this effect, R3F maintains more stable and physically-consistent reconstructions on foreground compared to prior methods, demonstrating its robustness in challenging real-world settings.

### A.7 USE OF LARGE LANGUAGE MODELS (LLMS)

We used large language models (LLMs) solely as a writing assist tool. In particular, we used LLMs to polish the readability of our text and to refine LaTeX commands. The research ideas, experimental design, and analysis were entirely conducted by the authors without LLM involvement.

### A.8 AUTHOR STATEMENT

The authors confirm that they bear full responsibility for any violations of rights related to the objects and data used in this work. All objects utilized in the dataset were either sourced from publicly available repositories with appropriate licensing or created by the authors. The data licenses are documented in Appendix A.4, and any modifications to the sourced objects were performed in compliance with the respective licenses. The authors ensure that all data used adhere to the specified licenses and terms of use.

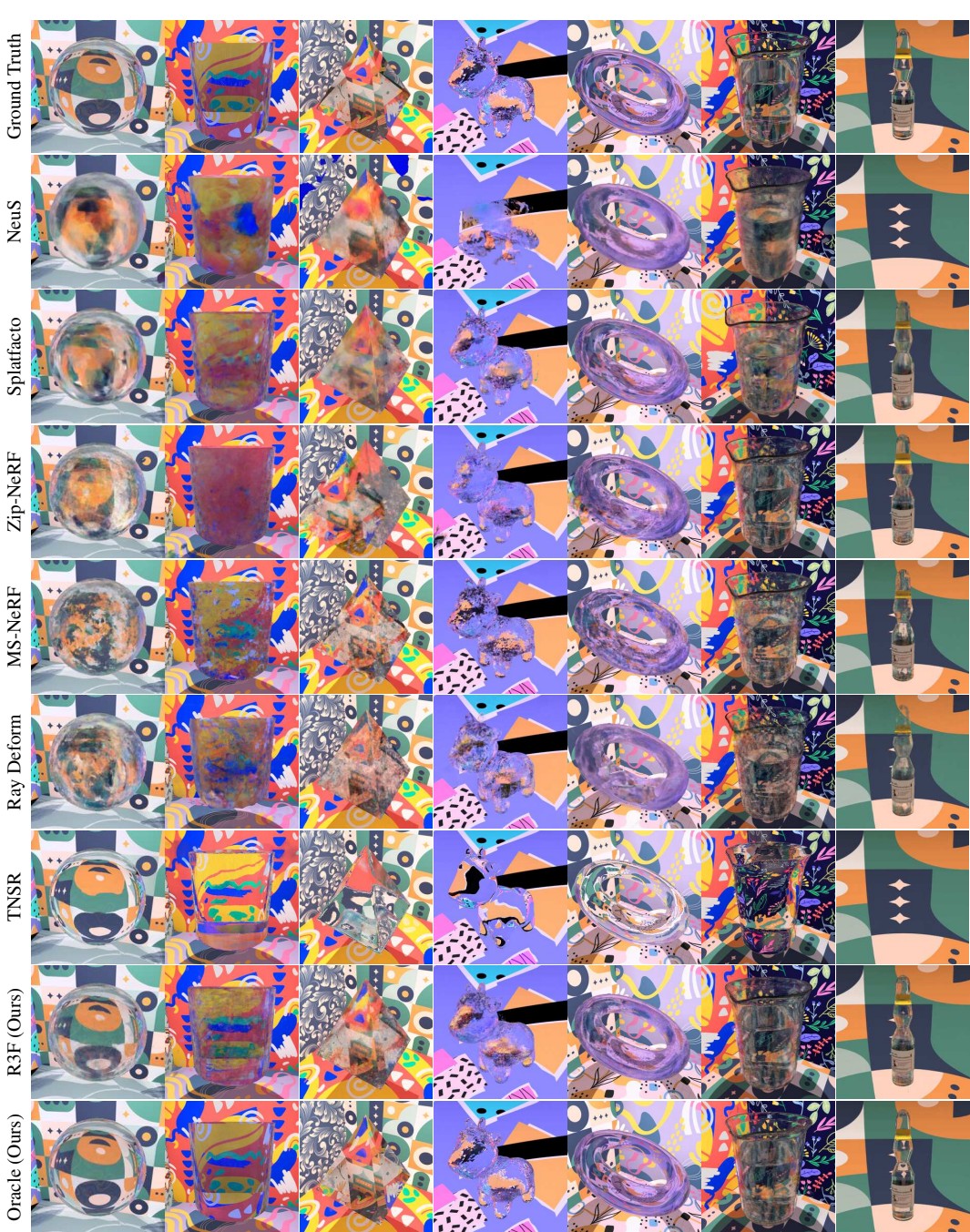

Figure 6: Qualitative comparison of novel view synthesis results on scenes with patterned cube and patterned sphere backgrounds using NeuS (Wang et al., 2021), Splatfacto (Tancik et al., 2023), Zip-NeRF (Barron et al., 2023), MS-NeRF (Yin et al., 2023), Ray Deformation Network (Deng et al., 2024b), TNSR (Deng et al., 2024a), R3F, and Oracle. R3F and Oracle produce more accurate renderings, especially in scenes involving multiple refractions and total internal reflection, where other methods often fail.

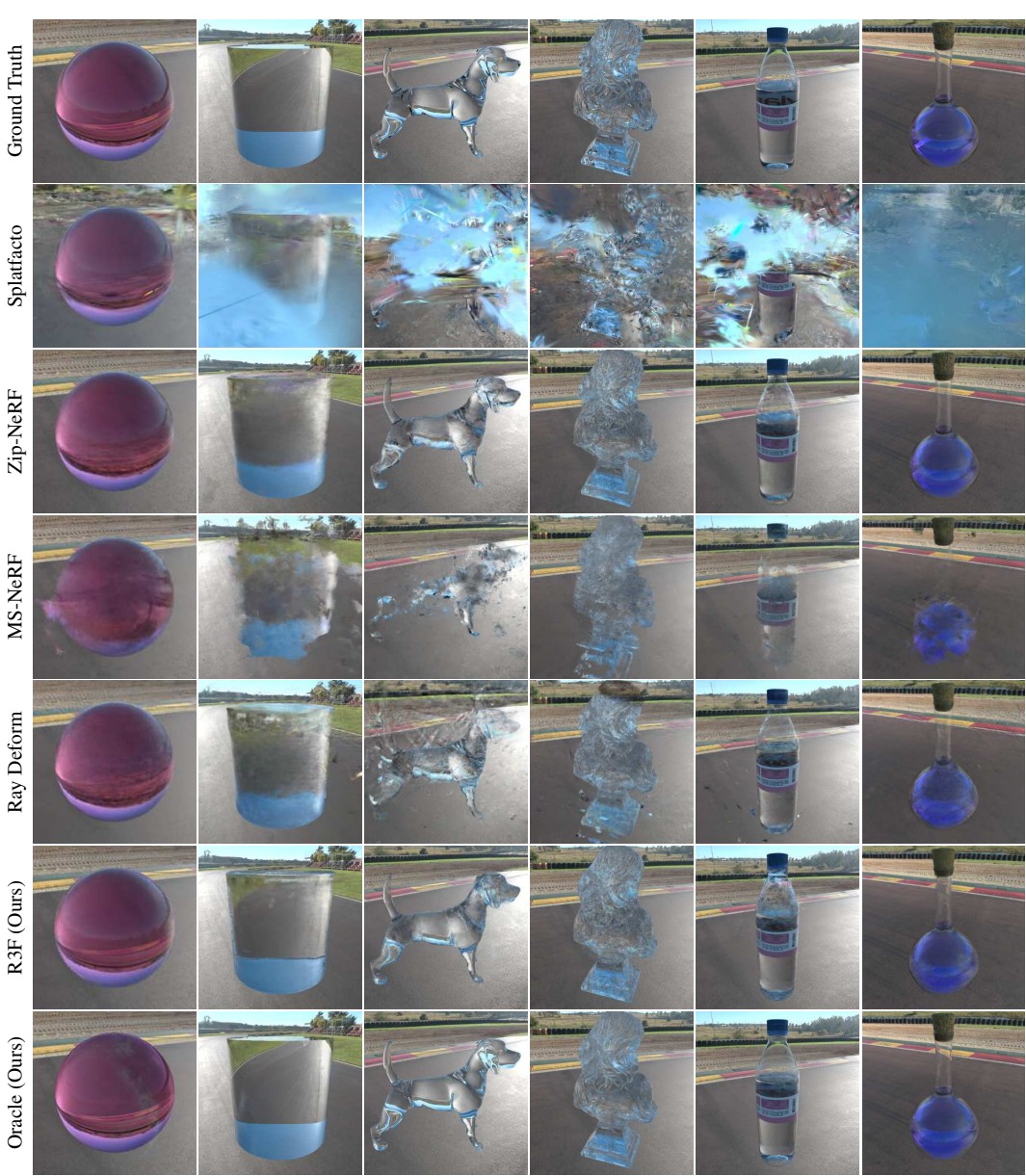

Figure 7: Qualitative comparison of novel view synthesis on scenes with HDR environment map backgrounds, using Splatfacto (Tancik et al., 2023), Zip-NeRF (Barron et al., 2023), MS-NeRF (Yin et al., 2023), Ray Deformation Network (Deng et al., 2024b), R3F (Ours), and Oracle. This background type is generally less challenging than the patterned ones. Most methods perform well on simple geometries, but their results vary significantly on complex shapes, where R3F and Oracle remain relatively robust.

