# OpenReview forum: "RefRef: A Dataset and Benchmark for Reconstructing Refractive and Reflective Objects"
_ICLR.cc/2026/Conference — ICLR 2026 Conference Withdrawn Submission_

### Official Review · Reviewer_J69D · 2025-10-27

**Soundness:** 3
**Presentation:** 3
**Contribution:** 2
**Rating:** 6
**Confidence:** 5

**Summary:**

The authors present a dataset and benchmark specifically designed to assess the performance of reflective and refractive objects. These objects pose significant challenges, making the establishment of an appropriate test set of utmost importance. To illustrate the intricacies of the task, the authors propose two baseline methods. One utilizes ground truth geometry and IOR information, while the other relaxes the geometry constraint by estimating a convex hull from masked images. Even with the availability of ground truth information, the method still fails to produce flawless reconstructions, underscoring the inherent difficulties in this task.

**Strengths:**

- It is intriguing to observe that even with access to GT geometry and IOR, the results still fall short of perfection, underscoring the inherent challenges involved.
- Although the underlying task remains unsolved, and even with access to elements from GT, the results are not yet fully satisfactory. Nevertheless, it is beneficial to formalize the issue within the community and establish a benchmark.
- The creative use of mirrored and angled R’s to indicate whether a reflected or refracted ray is employed enhances readability.
- The three stages of the dataset, comprising single materials (convex and non-convex) and multi-material non-convex, present an intriguing progression of challenges.

**Weaknesses:**

- I propose the inclusion of an additional category in the dataset, akin to numerous test scenes in computer graphics, featuring objects enclosed within glass surfaces, such as Lambertian torus within a glass cube.
- I believe it would be beneficial to evaluate Mitsuba in this context. The Wenzel Jakob team has dedicated substantial effort to solving this issue from a graphics perspective, even optimizing glass surfaces for projection rendering.
- Although formalizing a dataset is beneficial, it is synthetic and the proposed method offers limited insights.

**Questions:**

My main questions are related to the dataset: are the real scenes also be released? And is there an intention to extending it for mixed materials (See above in Weakness section)?

Also is there a reasoning for the degraded background modeling compared to that produced by ZipNeRF or other methods in the real data (Teaser)?

---

### Official Review · Reviewer_rtV6 · 2025-11-02

**Soundness:** 1
**Presentation:** 3
**Contribution:** 2
**Rating:** 2
**Confidence:** 5

**Summary:**

This paper contributes a dataset of reflective and refractive objects, along with two methods for nerf-ing such scenes. The first method is a theoretical upperbound where the GT shape and refractive indices are known, while the second uses (modern) visual hull to get the shape and TNSR to get the refractive index, and then repeats what the oracle method does. The oracle results look flawless to me mostly, so the takeaway point for me at least is nerf can render such objects well if we know the shape and refractive index.

The dataset itself has 50 synthetic objects in front of 3 backgrounds, in total making hence 150 scenes. The authors also captured some real scenes to validate the data and models. The objects are mostly of unsurprising shapes such as cube, sphere, cup, cat, etc.

**Strengths:**

Nice presentation that made the paper easy to follow.

I like the oracle upperbound to anchor the performance that the actual model R3F achieved. This is both helpful and thoughtful.

I also like how physics is wired in for the oracle method, disentangling rendering capability of nerf from reconstruction capability.

**Weaknesses:**

The actual R3F method is overly simplistic, basically differing from the (impractical) oracle in just some preprocessing steps such as getting the shape from modern visual hull. The preprocessing, as we all know, is very flaky. The method applies some post-hoc smoothing but then later doesn’t optimize the shape. This feels like a major drawback.

Even a bigger problem is the refractive index, which the authors just glossed over. It’s extremely hard to get that for real scenes with multiple materials.

These two points make R3F impractical. I was hoping for a visual hull initialization but then later optimization for better shape and estimating the refractive index. Not seeing either.

**Questions:**

Why not optimize the shape starting from the visual hull init?

How does one go about the refractive index for R3F?

Given how simplistic the shapes in the dataset are, how do people use this dataset for real-world applications?

---

### Official Review · Reviewer_GNMo · 2025-11-03

**Soundness:** 2
**Presentation:** 2
**Contribution:** 2
**Rating:** 4
**Confidence:** 5

**Summary:**

The submission introduces RefRef, a dataset and benchmark for novel view synthesis and 3D reconstruction of refractive + reflective objects. The dataset contains 50 objects × 3 backgrounds = 150 synthetic scenes plus one real scene, and objects are stratified by geometric/material difficulty: (i) single-material convex, (ii) single-material non-convex, (iii) non-convex multi-material (including tinted and bounded/unbounded setups). On top of the dataset, the authors build (1) an oracle renderer that assumes access to GT geometry and refractive indices and plugs this into a Zip-NeRF-style proposal-and-rendering pipeline, supporting up to 10 refractions + 1 explicit reflection + Fresnel mixing + TIR; and (2) a practical variant, R3F, that replaces GT geometry with a visual-hull-based reconstruction plus mesh smoothing, then reuses (most of) the oracle’s light-path logic. The paper then benchmarks several baselines on RefRef (Zip-NeRF, MS-NeRF, Ray Deformation Networks, TNSR, NeuS, Splatfacto, RoseNeRF) and shows a large gap to the GT-driven oracle, especially on geometry (DMAE) and on scenes with multiple refractions or total internal reflection.

**Strengths:**

* Taking a strong modern field (Zip-NeRF) and making it walk piecewise-linear, GT-informed paths (with TIR, first-surface reflection, and a Fresnel combination) is a good upper bound to have.
* The paper points out that Mip-NeRF360-style distortion loss tends to drive translucent regions to zero because it prefers single peaked weights, which is wrong for glass.

**Weaknesses:**

The core issue is positioning versus NeRRF [A] (i.e. earlier work that already did controlled, Blender-Cycles, multi-background, refractive+specular, geometry-aware rendering for transparent objects).

* Regarding dataset, NeRRF already did (1) Blender Cycles, physically correct rendering; (2) transparent + specular variants; (3) multiple backgrounds / HDRIs; (4) multi-view with known poses;
* Regarding Oracle, NeRRF already explored the idea: if you give me the geometry + material, I can trace the physically plausible light path and the problem becomes easy.
* Regarding method, the method is also similar to NeRRF. The pipeline logic is the same shape: (1) get object masks; (2) reconstruct a clean enough mesh; (3) run physically motivated refraction/reflection on the estimated geometry; (4) render. NeRRF did this with DMTet + progressive encoding + eikonal to get a closed and smooth surface.
* Regarding benchmark, NeRRF likewise observed: (1) straight-ray NeRFs collapse on strong refraction; (2) purely specular methods don’t help with refraction; (3) small geometry errors destroy the whole light path.

[A] Nerrf: 3d reconstruction and view synthesis for transparent and specular objects with neural refractive-reflective fields

**Questions:**

Could authors provide a systematic comparison to NeRRF?

---

### Official Review · Reviewer_gEPf · 2025-11-03

**Soundness:** 3
**Presentation:** 3
**Contribution:** 2
**Rating:** 4
**Confidence:** 2

**Summary:**

The paper proposes a method and benchmark for reconstructing refractive and reflective object-centric scenes in a NeRF setting. Specifically,
- The paper introduces a dataset consisting of a few dozen scenes exclusively of reflective and refractive materials, as a contribution by itself as well as to evaluate the proposed method as well as baseline methods;
- A method that models reflecting and refracting light paths, under two assumptions: given oracle geomety and refraction index, VS input convex geometry ony;
- A benchmark using the proposed dataset against baseline methods.

**Strengths:**

- The proposed dataset consists of renderings of considerable amount (150) of synthetic scenes of varying combinations of material (single/mixed) and geometry (convex/non-convex), as well as camera and background settings, which is beneficial comprehensive evaluation on this particular task;
- The proposed paper is evaluated on settings (the proposed method under oracle input/convex-geometry input) as well as compared against baseline methods on different types of scenes (convex/ono-convex, complex geometry, complex material). Discussion of the view synthesis and geoemtry estimtaiton results is also provided on the strength and limitations of the method under various conditions;
- The paper is generally well-written with detailed mathematical formulation of sampling along reflective/refractive paths and illustrative figures;
- Novel design choices in the method are shown to lead to proformance gains, including a modified distortion losses which encourages density inside transluscent objects versus baseline loss from Mip-NeRF 360;

**Weaknesses:**

- Insufficient discussion is provided on the most competitive method i.e. TNSR, considering TNSR achieve comparable results in some test scenes or scenarios, and employ similar ray tracing formalution during optimization. Please see Questions sections for additional discussion and results to be included.

- Ablation and discussion is provided on the limitations of the proposed method, but a few key questions have to be unanswered to fully access the method. Please see Questions.

**Questions:**

(1) How is the method in TNSR differentiate from the proposed method in this paper, given they both attempt to model light transport with reflective and refractive materials? Without the discussion it is difficult to gauge the theoretical novelty of the proposed method vs TNSR;

(2) In which cases and why is TNSR achieves competitive results in some test scenes (e.g. the rotating glass ball in the Supp. video)?

(3) What if input convex geometry or oracle input is provided to TNSR, and other methods (if applicable)? In this case how does the proposed methods compare to those baseline endowed with input geomety or GT geometry/material information?

(4) (Fig. 5 and L459) Why is the proposed method struggles with complex geometry (even in oracle setting with perfect geometry)? Are there any ideas on what need to be done to address the limitation?

---

### Note · Authors · 2025-11-13

I have read and agree with the venue's withdrawal policy on behalf of myself and my co-authors.